

# The prognostic value of preoperative serum CA724 for CEA-normal colorectal cancer patients

Jiaan Kuang[1,*], Yizhen Gong[1,*], Hailun Xie[1], Ling Yan[1], Shizhen Huang[1], Feng Gao[1], Shuangyi Tang[2] and Jialiang Gan[1]

[1] Department of Colorectal Anal Surgery, The First Affiliated Hospital of Guangxi Medical University, Nanning, Guangxi, China
[2] Department of Pharmacy, The First Affiliated Hospital of Guangxi Medical University, Nanning, Guangxi, China
* These authors contributed equally to this work.

Corresponding author
Jialiang Gan, gjl5172@163.com

## ABSTRACT

**Background and Purpose:** There had been no recognized serum tumor marker to predict the prognosis of colorectal cancer (CRC) patients with normal preoperative serum carcinoembryonic antigen (CEA) levels. The purpose of this study was to determine whether preoperative serum carbohydrate antigen 724 (CA724) was of predictive function for the prognosis of CRC patients with normal CEA levels.
**Methods:** The medical records of 295 CRC patients with normal CEA levels who underwent surgery at the Department of Colorectal Anal Surgery of the First Affiliated Hospital of Guangxi Medical University (Guangxi, China) between September 2012 and September 2014 were retrospectively reviewed. The Chi-square test was used to test the correlation between preoperative serum CA724 levels and clinical features. Kaplan–Meier curves were conducted to calculate the overall survival (OS) rate and disease-free survival (DFS) of patients. Cox regression analysis was applied to conduct univariate and multivariate analysis of the following four preoperative serum tumor makers namely CA724, carbohydrate antigen 199 (CA199), carcinoembryonic antigen 125 (CA125), carcinoembryonic antigen 242 (CA242) and clinical features. Nomograms for prognostic parameter of OS and DFS were developed using R v3.2.5.
**Results:** In the Chi-square test, only pathological node stage (pN stage) ($X^2 = 14.514$, $P = 0.001$) and differentiation ($X^2 = 10.712$, $P = 0.001$) were associated with serum CA724 levels. In the Kaplan–Meier analysis, the results revealed that the OS and DFS in patients with high CA724 was poorer than those with normal. In the multivariate Cox regression analysis of OS and DFS, only pT stage, pN stage, metastasis and serum CA724 were independent prognostic risk factors for CRC patients with normal CEA levels.
**Conclusion:** Preoperative serum CA724 might serve as a potential prognostic factor for CRC patients with normal serum CEA levels.

## INTRODUCTION

Colorectal cancer (CRC) is the third most common malignancy and the second most common cause of cancer-related mortality worldwide (*Bray et al., 2018*). In China, CRC is the fifth most common malignancy and the incidence of this disease continues to increase gradually (*Chen et al., 2016b*). Surgical resection is still a mainstay of curative treatment for CRC (*Benson et al., 2018*). However, survival after radical resection in some patients was poor because of recurrence and metastasis. Some markers to predict prognosis had been discovered for screening of risk for recurrence or metastasis (*Feng et al., 2018*; *Nakamura & Yoshino, 2018*). But their performance on clinic appliance was imperfect and some testing was only implemented in more comprehensive medical institutions due to the technology and expense. Pathologic stage is an important prognostic factor, but its adequacy had been recently questioned (*Lea et al., 2014*; *Li et al., 2014*). Therefore, it was essential to identify other markers to predict the prognosis of CRC patients.

Serum tumor markers are produced and released during tumorigenesis and the levels of serum tumor markers may be indicative of tumor growth (*Jo et al., 2013*). Many valuable studies had suggested that preoperative serum carcinoembryonic antigen (CEA) levels could be used as independent risk factors to estimate the prognosis of CRC patients (*Chen et al., 2017*; *Kozman et al., 2018*; *Peng et al., 2015*). However, the high rate of serum CEA in CRC patients was less than 50% in some clinical trials (*Huang et al., 2017*, *2018*; *Shinkins et al., 2017*). Furthermore, at present, there was no recognized reliable serum tumor marker for prognostic use in serum CEA-normal CRC patients. Carbohydrate antigen 724 (CA724) was the most correlative serum tumor biomarker for gastric cancer (*Chen et al., 2012*). It had been recently reported that serum CA724 was closely correlated to tumor staging of CRC patients (*Zhu et al., 2014*). But, the relationship between serum CA724 levels and prognosis of CRC remained unclear, especially in the patients with normal CEA levels. Therefore, the aim of the present study was to determine the utility of preoperative serum CA724 levels to predict the prognosis of CRC patients with normal CEA levels.

## METHODS

### Study population

The medical records of the CRC patients who underwent surgery at the Department of Colorectal Anal Surgery, the First Affiliated Hospital of Guangxi Medical University of between September 2012 and September 2014 were collected. The cases were chosen according to the following inclusion criteria: (1) histopathological diagnosis of colon or rectal cancer; (2) primary tumor that can be radically removed; (3) complete clinical pathology report and postoperative follow-up data; (4) complete data of preoperative serum oncologic markers. The cases were excluded according to the following criteria: (1) lack of histopathological confirmation of colon or rectal cancer; (2) preoperative neoadjuvant treatment; (3) death cause other than CRC-related; (4) other simultaneous or heterogenic malignant tumors (such as lymphoma, leiomyosarcoma, stromal tumor,

melanoma, etc.); (5) no signature on the informed consent form for disposal of biological specimens.

## Collection of serum tumor marker and clinicopathological characteristics

Preoperative fasting venous blood (four mL) was collected on the second day after the admission of the patients and centrifuged within 1 h after collection. Serum that had not be analyzed within 6 h was stored in a refrigerator at −20 °C. The serum levels of CEA, CA724, carbohydrate antigen 199 (CA199), carcinoembryonic antigen 125 (CA125) and carcinoembryonic antigen 242 (CA242) were measured using a chemiluminescence immunoassay and the Elecsys 2010 Immunoassay Analyzer (Roche Diagnostics, Risch-Rotkreuz, Switzerland). The following concentrations were considered to indicate high expression: CEA > 5 ng/mL, CA199 > 37 U/mL, CA724 > 5.7 U/mL, CA242 > 20.0 U/mL and CA125 > 35.0 U/mL. These clinicopathological characteristics were collected in this study including age, gender, pathological tumor stage (pT stage), pathological node stage (pN stage), metastasis, tumor location, venous invasion, perineural invasion, pathologic type, differentiation, postoperative chemoradiotherapy. The patients were staged according to the American Joint Committee on Cancer staging manual (seventh edition). Radical operation was colorectal resection plus regional lymph node dissection (*Cohen, 1991*; *Nascimbeni et al., 2002*; *West et al., 2009*).

## Survival follow-up

Patients were followed-up by telephone or outpatient once every 3 months for the first 2 years after surgery and then every 6 months thereafter. Follow-up examinations included measurements of serum tumor marker levels, chest, abdomen, pelvic imaging by computed tomography and electronic colonoscopy. The follow-up period was from the time of the discharge to the time of death or 1st July 2019. The censored data of this study was defined as the data of the patients who were survival at the final follow-up deadline. Overall survival (OS) was defined as the time from resection of colorectal cancer to death from colorectal cancer or the censored time. Disease-free survival (DFS) was defined as the time from diagnosis to the first recurrence.

## Statistical analyses

All statistical analyses were conducted using IBM SPSS Statistics for Windows, version 24.0 (IBM Corp, Armonk, NY, USA). The Chi-square test was used to identify associations between serum CA724 levels and clinicopathological features. We applied Kaplan–Meier curve to estimate the survival curve of OS and DFS, which was compared with the log-rank test. Univariate, multivariate and subgroup survival analysis were performed by using Cox proportional hazards model. All factors that were identified as significant associated with OS and DFS by univariate analysis were subjected to multivariate Cox proportional hazards analysis. Nomograms for predicting OS and DFS were developed using R v3.2.5. A probability (*P*) value of <0.05 was considered statistically significant. The confidence level of confidence intervals at the 95% (95% CI) was used in this study

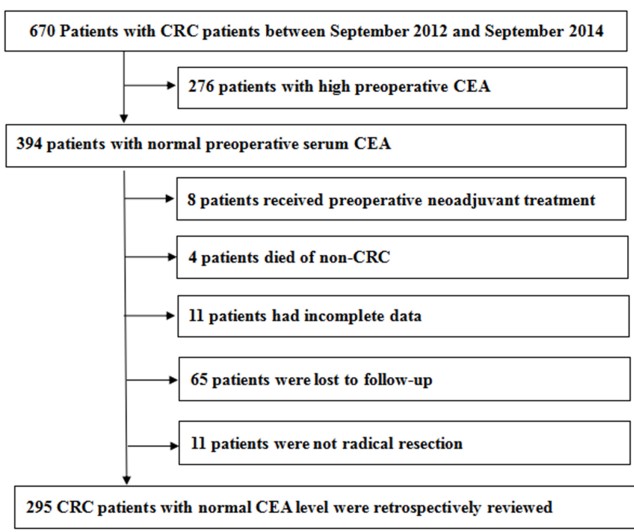

**Figure 1 The process of case inclusion and exclusion in this study.**

## Ethics statement

The study protocol was approved by the Hospital Ethics Committee of the First Affiliated Hospital of Guangxi Medical University, Guangxi, China-Approval number: 2018 (KY-E-086). All patients signed informed consent for collection and analysis of biological specimens. This study was conducted in accordance with the Declaration of Helsinki.

## RESULTS

### Clinical characteristics

We collected medical records of 670 CRC patients. The process of case inclusion and exclusion in this study was shown in Fig. 1. There were 394 patients whose preoperative serum CEA was normal. We excluded patients who had received preoperative adjuvant therapy, incomplete data, non-colorectal cancer deaths and not radical resection. There were 65 patients who were lost to follow-up. Finally, the medical records of 295 CRC patients with normal CEA level were retrospectively reviewed. The median follow-up time of OS was 65 months and DFS was 64 months.

The characteristics of these patients were displayed in Table 1. Among those 295 patients, there were 223 having tumors in the left side and 72 in the right side. The pathological stages of the tumors were as follows: 98 as T1-2, 197 as T3-4 and 186 as N0, 71 as N1, and 38 as N2. The tumors of 261 patients were well/moderately differentiated and those of 34 were poorly differentiated. Venous invasion or perineural invasion was detected in 48 patients and metastasis was detected in 11 patients.

### The correlation between preoperative serum CA724 levels and clinical features

Chi-square test was used to test the correlation between preoperative serum CA724 levels and clinical features (Table 1), which included age, gender, tumor location, pT stage,

**Table 1  Comparison of baseline clinicopathological characteristics based on CA724 in CEA-normal CRC patients.**

| Feature | Case | CA724 | | $\chi^2$ | P |
|---|---|---|---|---|---|
| | | Normal (242) | High (53) | | |
| Gender | | | | 0.026 | 0.872 |
| Male | 181 (61.36%) | 149 (61.6%) | 32 (60.4%) | | |
| Female | 114 (38.64%) | 93 (38.4%) | 21 (39.6%) | | |
| Age (Year) | | | | 3.424 | 0.064 |
| ≤65 | 161 (54.58%) | 126 (52.1%) | 35 (66.0%) | | |
| >65 | 134 (45.42%) | 116 (47.9%) | 18 (34.0%) | | |
| pT stage | | | | 0.594 | 0.441 |
| T1–2 | 98 (33.22%) | 78 (32.2%) | 20 (37.7%) | | |
| T3–4 | 197 (66.78%) | 164 (67.8%) | 33 (62.3%) | | |
| pN stage | | | | 14.514 | 0.001 |
| N0 | 186 (63.05%) | 156 (64.5%) | 30 (56.6%) | | |
| N1 | 71 (24.07%) | 63 (26.0%) | 8 (15.1%) | | |
| N2 | 38 (12.88%) | 23 (9.5%) | 15 (28.3%) | | |
| Metastasis | | | | 0.611 | 0.435 |
| No | 284 (96.27%) | 232 (95.9%) | 52 (98.1%) | | |
| Yes | 11 (3.73%) | 10 (4.1%) | 1 (1.9%) | | |
| Tumor location | | | | 0.001 | 0.982 |
| Left side | 223 (75.6%) | 183 (75.6%) | 40 (75.5%) | | |
| Right side | 72 (24.4%) | 59 (24.4%) | 13 (24.5%) | | |
| Perineural/vascular invasion | | | | 3.233 | 0.072 |
| No | 247 (83.73%) | 207 (85.5%) | 40 (75.5%) | | |
| Yes | 48 (16.27%) | 35 (14.5%) | 13 (24.5%) | | |
| Pathological type | | | | 2.101 | 0.350 |
| Protrude type | 57 (19.32%) | 43 (17.8%) | 14 (26.4%) | | |
| Infiltrating type | 35 (11.86%) | 29 (12.0%) | 6 (11.3%) | | |
| Ulcerative type | 203 (68.82%) | 170 (70.2%) | 33 (62.3%) | | |
| Differentiation | | | | 10.712 | 0.001 |
| Poor | 34 (11.53%) | 21 (8.7%) | 13 (24.5%) | | |
| Medium/High | 261 (88.47%) | 221 (91.3%) | 40 (75.5%) | | |

**Note:**
CEA, carcinoembryonic antigen; CRC, colorectal cancer; pT stage, pathological tumor stage; pN stage, pathological node stage.

pN stage, differentiation, venous invasion, perineural invasion, metastasis and pathologic type. Among these clinicopathological features, only pathological node stage (pN stage) ($X^2$ = 14.514, $P$ = 0.001) and differentiation ($X^2$ = 10.712, $P$ = 0.001) were associated with serum CA724 levels.

## Prognostic values of serum tumor markers

In the Kaplan–Meier analysis, the results revealed that the OS and DFS of the patients with high CA724 was poorer than those with normal (Figs. 2B and 2F), but there was no

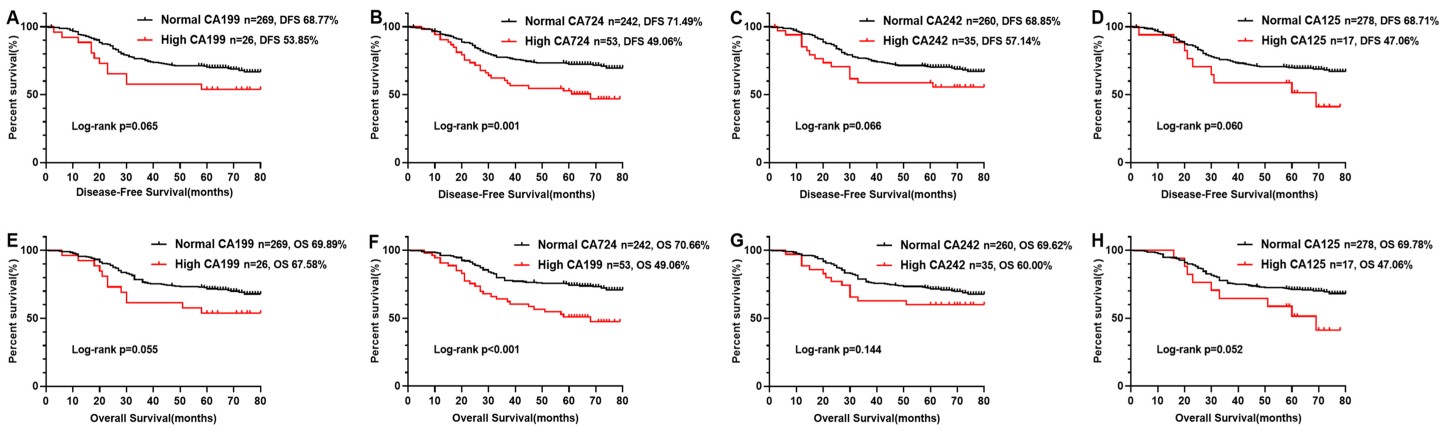

**Figure 2 Comparison of OS rate and DFS rate of CRC patients with normal CEA levels according to CA724, CA199, CA242, CA125.** The results of the Kaplan–Meier analysis revealed that the OS rate and DFS rate of the patients with high CA724 was poorer than those with normal (B, F), but there was no significant difference between CA199, CA242, CA125 high patients and CA199 (A, E), CA242 (C, G), CA125 (D, H) normal patients (*P* < 0.05).

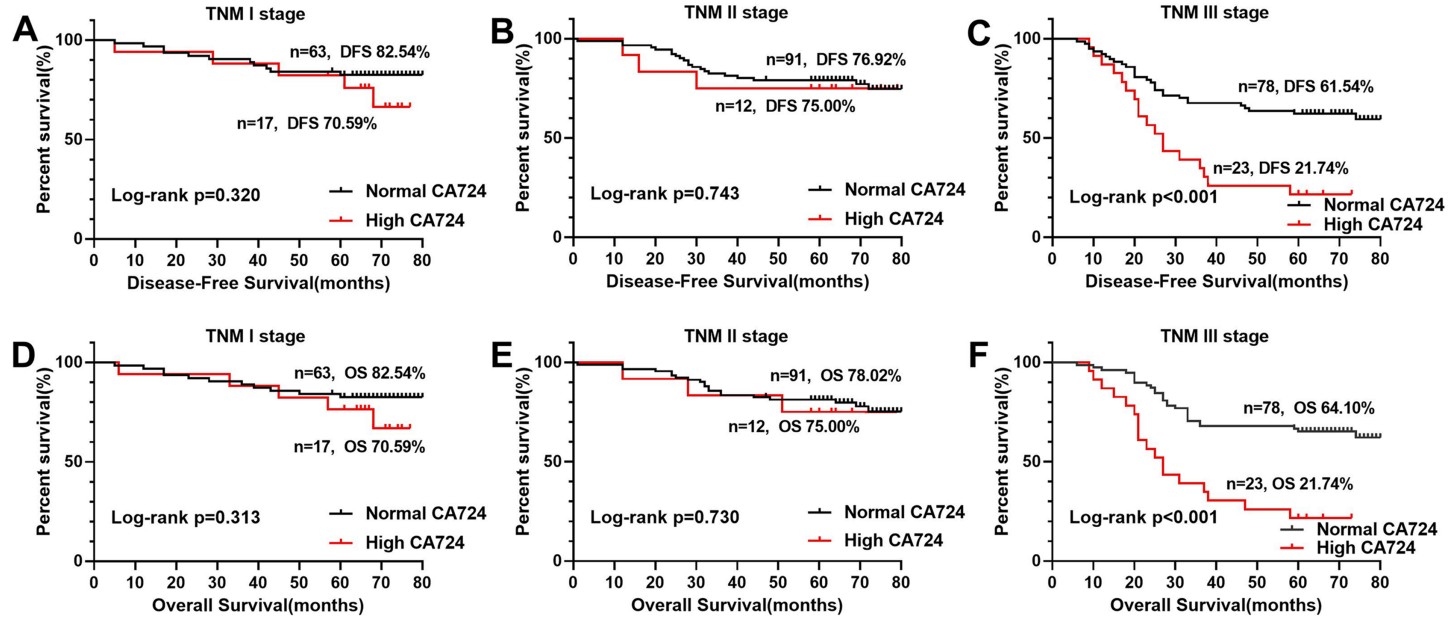

**Figure 3 Comparison of OS and DFS of TMN stage according to CA724.** There was significant difference between CA724 high patients and those with normal in the Kaplan–Meier analysis of TNM stage III (C, F), but there was no significant difference at TNM stage I ( A, D) and TNM stage II (B, E) (*P* < 0.05).

significant difference between CA199, CA242, CA125 high patients and CA199, CA242, CA125 normal patients (Figs. 2A, 2C–2E, 2G and 2H). And there was significant difference between CA724 high patients and those with normal in the Kaplan–Meier analysis of TNM stage III (Fig. 3). As shown in Tables 2 and 3, pT stage, pN stage, metastasis, vascular or neurological invasion, differentiation and serum levels of CA724 were significantly correlated with OS and DFS by univariate analysis. In the multivariate Cox regression

**Table 2 Univariate and multivariate overall survival analyses of clinicopathological covariates in CRC patients.**

| Feature | Overall survival | | | |
|---|---|---|---|---|
| | Univariate analysis | | Multivariate analysis | |
| | HR (95% CI) | *P* | HR (95% CI) | *P* |
| Gender | | 0.885 | | |
| Male | 1 | | | |
| Female | 0.970 [0.637–1.475] | | | |
| Age (Year) | | 0.161 | | |
| ≤65 | 1 | | | |
| >65 | 1.338 [0.891–2.011] | | | |
| pT stage | | 0.002 | | 0.039 |
| T1–2 | 1 | | 1 | |
| T3–4 | 2.230 [1.347–3.693] | | 1.761 [1.029–3.014] | |
| pN stage | | <0.001 | | <0.001 |
| N0 | 1 | | 1 | |
| N1 | 1.788 [1.084–2.949] | | 1.589 [0.943–2.676] | |
| N2 | 5.524 [3.391–8.998] | | 3.995 [2.304–6.927] | |
| Metastasis | | <0.001 | | 0.001 |
| No | 1 | | 1 | |
| Yes | 4.078 [1.964–8.467] | | 3.509 [1.624–7.582] | |
| Tumor location | | 0.419 | | |
| Left side | 1 | | | |
| Right side | 1.208 [0.764–1.911] | | | |
| Perineural/vascular invasion | | 0.005 | | 0.179 |
| No | 1 | | 1 | |
| Yes | 1.960 [1.223–3.141] | | 1.418 [0.852–2.358] | |
| Pathological type | | 0.697 | | |
| Protrude type | 1 | | | |
| Infiltrating type | 1.354 [0.634–2.894] | | | |
| Ulcerative type | 1.234 [0.705–2.162] | | | |
| Differentiation | | 0.019 | | 0.257 |
| Poor | 1 | | 1 | |
| Medium/high | 0.523 [0.305–0.898] | | 1.427 [0.772–2.639] | |
| Postoperative chemoradiotherapy | | 0.786 | | |
| No | 1 | | | |
| Yes | 1.058 [0.704–1.591] | | | |
| CA199 | | 0.060 | | |
| Normal | 1 | | | |
| High | 1.790 [0.976–3.283] | | | |
| CA724 | | <0.001 | | 0.001 |
| Normal | 1 | | 1 | |
| High | 2.271 [1.450–3.557] | | 2.261 [1.379–3.708] | |

(Continued)

| Feature | Overall survival | | | |
| --- | --- | --- | --- | --- |
| | Univariate analysis | | Multivariate analysis | |
| | HR (95% CI) | P | HR (95% CI) | P |
| CA242 | | 0.149 | | |
|   Normal | 1 | | | |
|   High | 1.520 [0.861–2.685] | | | |
| CA125 | | 0.058 | | |
|   Normal | 1 | | | |
|   High | 1.945 [0.977–3.868] | | | |

Note:
CEA, carcinoembryonic antigen; CRC, colorectal cancer; pT stage, pathological tumor stage; pN stage, pathological node stage.

analysis of OS, the result indicated that only pT stage ($P = 0.039$), pN stage ($P < 0.001$), metastasis ($P = 0.001$) and serum CA724 ($P = 0.001$) were independent prognostic factors for CRC patients with normal serum CEA levels. In the multivariate Cox regression analysis of DFS, the result showed the same independent prognostic factors namely pT stage ($P = 0.036$), pN stage ($P < 0.001$), metastasis ($P = 0.003$) and serum CA724 ($P = 0.005$) for CRC patients with normal serum CEA levels.

## Subgroup analysis of clinical features

The results of multivariate Cox proportional hazards analysis suggested that CA724 was an independent prognostic risk factor for CRC patients with normal CEA. In order to further assess the prognostic value of CA724 in CEA normal patients, CA724 and other factors that identified significant association with DFS and OS by univariate analysis were subjected to multivariate Cox proportional hazards analysis for subgroup analysis of each clinical feature. The statistical results of CA724 in each subgroup were demonstrated in Fig. 4. In the analysis of 11 subgroups, there were statistical significances with DFS in the subgroups of female, age group (>65), T3–T4 stage, no metastasis, right side, no venous invasion and perineural invasion, well/moderate differentiation, no postoperative chemoradiotherapy, while there were statistical significances with OS in the subgroups of gender, age group (>65), T3–T4 stage, N2 stage, tumor location, venous invasion and perineural invasion, protrude type, ulcerative typ, well/moderate differentiation, postoperative chemoradiotherapy.

## Nomogram for predicting survival outcomes

Two nomograms were established to evaluate the relationship between CA724 and medical rank in CRC patients with normal CEA (Fig. 5). Based on the results of multivariate Cox proportional hazards analysis, only CA724, pT stage, pN stage and metastasis were entered the risk model. The survival rate of 1–5 years could be predicted and CA724 played an important role in the nomogram.

**Table 3 Univariate and multivariate disease-free survival analyses of clinicopathological covariates in CRC patients.**

| Feature | Disease-free survival | | | |
| --- | --- | --- | --- | --- |
| | Univariate analysis | | Multivariate analysis | |
| | HR (95% CI) | *P* | HR (95% CI) | *P* |
| Gender | | 0.898 | | |
| Male | 1 | | | |
| Female | 0.974 [0.645–1.469] | | | |
| Age (Year) | | 0.403 | | |
| ≤65 | 1 | | | |
| >65 | 1.186 [0.795–1.770] | | | |
| pT stage | | 0.001 | | 0.036 |
| T1–2 | 1 | | 1 | |
| T3–4 | 2.244 [1.371–3.675] | | 1.751 [1.038–2.956] | |
| pN stage | | <0.001 | | <0.001 |
| N0 | 1 | | 1 | |
| N1 | 1.944 [1.195–3.161] | | 1.714 [1.031–2.849] | |
| N2 | 5.462 [3.361–8.874] | | 3.820 [2.192–6.660] | |
| Metastasis | | <0.001 | | 0.003 |
| No | 1 | | 1 | |
| Yes | 4.629 [2.231–9.604] | | 3.292 [1.514–7.154] | |
| Tumor location | | 0.245 | | |
| Left side | 1 | | | |
| Right side | 1.302 [0.835–2.033] | | | |
| Perineural/vascular invasion | | 0.009 | | 0.285 |
| No | 1 | | 1 | |
| Yes | 1.874 [1.172–2.996] | | 1.316 [0.795–2.178] | |
| Pathological type | | 0.793 | | |
| Protrude type | 1 | | | |
| Infiltrating type | 1.256 [0.594–2.655] | | | |
| Ulcerative type | 1.187 [0.688–2.046] | | | |
| Differentiation | | 0.023 | | 0.294 |
| Poor | 1 | | 1 | |
| Medium/high | 0.536 [0.313–0.918] | | 1.383 [0.754–2.536] | |
| Postoperative chemoradiotherapy | | 0.706 | | |
| No | 1 | | | |
| Yes | 1.080 [0.723–1.614] | | | |
| CA199 | | 0.069 | | |
| Normal | 1 | | | |
| High | 1.753 [0.957–3.210] | | | |
| CA724 | | 0.001 | | 0.005 |
| Normal | 1 | | 1 | |
| High | 2.097 [1.342–3.275] | | 2.028 [1.238–3.320] | |

(Continued)

| Feature | Disease-free survival | | | |
|---|---|---|---|---|
| | Univariate analysis | | Multivariate analysis | |
| | HR (95% CI) | P | HR (95% CI) | P |
| CA242 | | 0.070 | | |
| Normal | 1 | | | |
| High | 1.664 [0.959–2.889] | | | |
| CA125 | | 0.066 | | |
| Normal | 1 | | | |
| High | 1.906 [0.959–3.787] | | | |

**Note:**
CEA, carcinoembryonic antigen; CRC, colorectal cancer; pT stage, pathological tumor stage; pN stage, pathological node stage.

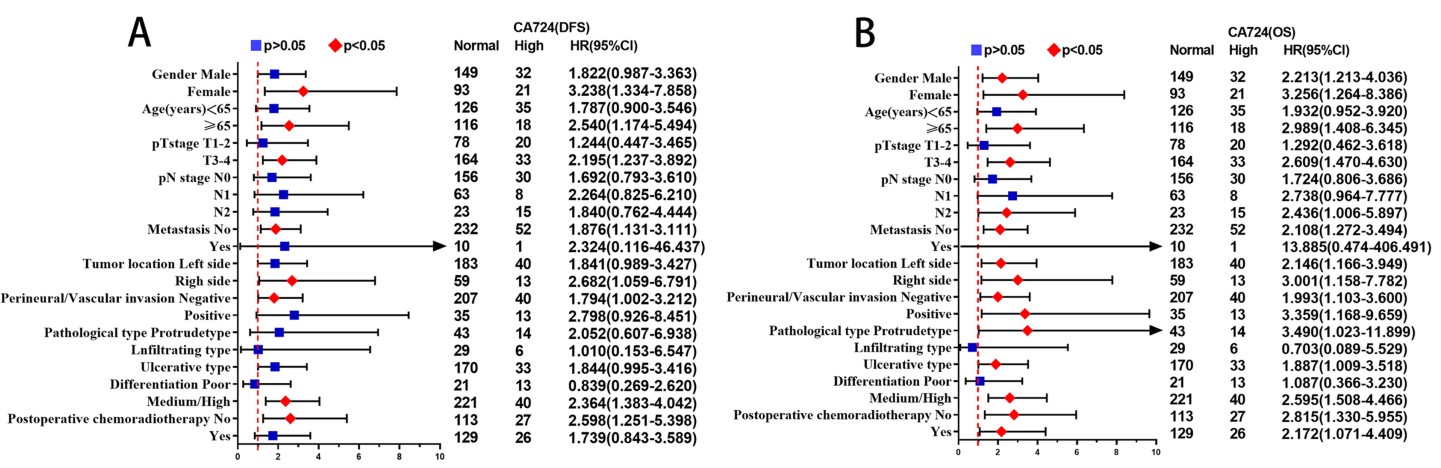

**Figure 4** **The statistical results of CA724 in each subgroup.** In the analysis of 11 subgroups, there were statistical significances with DFS (A) in the subgroups of female, age group (>65), T3–T4 stage, no metastasis, right side, no venous invasion and perineural invasion, well/moderate differentiation, no postoperative chemoradiotherapy. There were statistical significances with OS (B) in the subgroups of gender, age group (>65), T3–T4 stage, N2 stage, tumor location, venous invasion and perineural invasion, protrude type, ulcerative typ, well/moderate differentiation, postoperative chemoradiotherapy.

## DISCUSSION

More reliable markers were needed to predict the prognosis of CRC patients in a more sufficient manner. The measurement of serum tumor marker levels was a reliable, simple, effective and economical method to evaluate the prognosis of cancer patients (*Huang et al., 2016*; *Zou & Qian, 2014*). CEA, CA199, CA125, CA724, CA242 were commonly used serum biomarkers for preoperative diagnose, postoperative monitoring and efficacy evaluation in gastrointestinal malignancies. But, these serum tumor markers had no specificity for CRC. To the best of our knowledge, there was no a specific serum tumor marker to diagnose CRC despite of the fact that many researchers and scientists had poured great efforts in it. The combined use or supplementary application of the above known serum biomarkers was considered to be potentially effective and practical way to

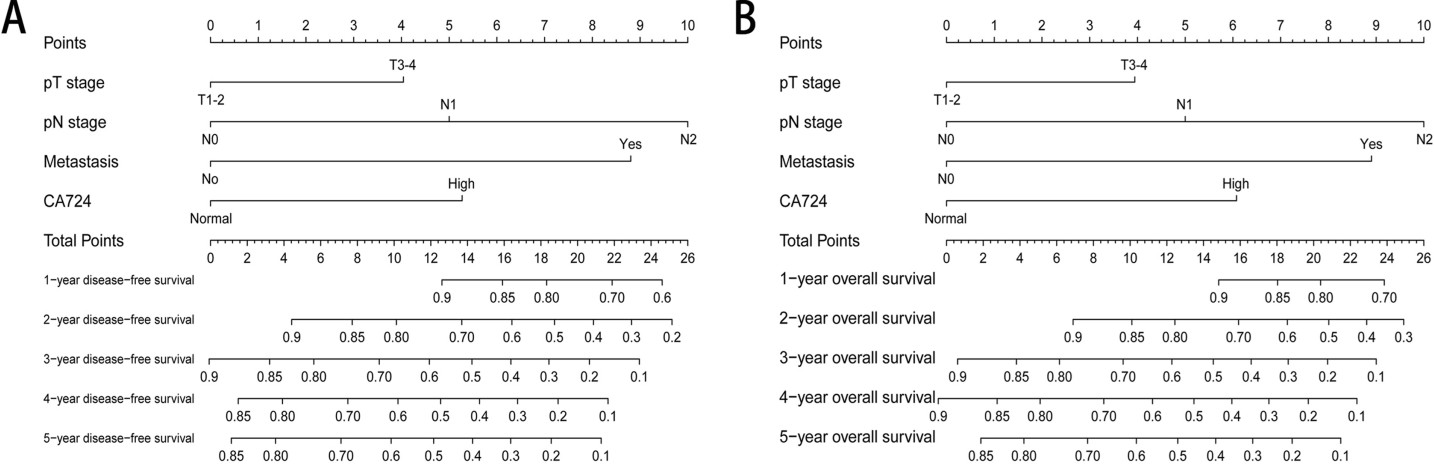

**Figure 5 The nomogram for predicting survival outcomes.** Two nomograms were established to evaluate the relationship between CA724 and medical rank in CRC patients with normal CEA. The survival rate of 1–5 years disease-free survival (A) and 1–5 years overall survival (B) could be predicted.

improve the high rate of CRC on diagnosis (*Kunizaki et al., 2016*; *Zhang et al., 2016*). CEA was an important serum marker for evaluating prognosis of CRC patients (*Lech et al., 2016*). However, the high rate of CEA was generally low. According to our retrospective analysis in large series of case investigation, the high rate of preoperative CEA was only 38.04% in 1969 CRC patients (*JiaLiang et al., 2013*), which suggested that most of the patients belonged to the preoperative CEA-normal CRC patients and would be lack of efficient serum biomarkers for CRC patients to monitor and follow-up after treatment. In this study, the results of multivariate analyses showed that among several serum tumor markers (CA125, CA199, CA724 and CA242), only CA724 was a statistically significant independent risk factor for the prognosis of CRC patients with normal preoperative serum CEA levels. Generally, the patients with a later TNM stage had a worse prognosis. In the statistically significant subgroups, the prognosis was poor for CA724-high patients and there was only significant difference between CA724 high patients and those with normal in the TNM stage III. We could hold that CA724 performed an important role in predicting prognosis of CEA-normal patients.

This study was the first to show the predictive value of preoperative serum CA724 levels for the prognosis of CRC patients with normal CEA levels. CA724 was a high molecular weight mucin-like glycoprotein that was upregulated in malignant tumors of the gastrointestinal tract, reproductive system and lung (*Liang et al., 2013*). The prognostic value of serum CA724 to detect the recurrence of gastric cancer was first established in the early 1990s (*Chen et al., 2014*). *Yanqing, Cheng & Ling (2018)* showed that the diagnostic sensitivity was low (0.50) but the specificity was high (0.86) for serum CA724 as a biomarker in the diagnosis of colorectal cancer. It indicated that CA724 still had a close relationship with colorectal cancer. *Liu, Zhu & Liu (2015)* showed that CA724 was a better predictor of the prognosis of unresectable pancreatic cancer relative to CA199 and CA125, which suggested that the production and release of CA724 might be similar to that in gastrointestinal malignancies.

In addition, serum CA724 levels and pN stage were independent prognostic factors of CRC, indicating that both were correlated to tumorigenesis. A study by Sun et al. (2017) showed that an elevation in serum CA724 levels suggested that tumor cells had metastasized through the lymphatic or other pathways and had colonized at an another location. The lymph system was a common metastatic pathway of gastrointestinal cancer (Li et al., 2013). Chen et al. (2016a) reported that serum CA724 might be associated with the overexpression of human epidermal growth factor receptor 2 (HER2). In some malignancies, HER2 was a targeted therapeutic marker. These studies demonstrated that CA724 played an important role in tumor development and might be valuable as a prognostic marker of CRC patients, especially in the CRC patients with normal CEA. This retrospective study was a single cohort study that based on the limited data available. NormalTherefore, it was necessary to expand the sample from self-validation, or further validate our point of view through horizontal multi-center data in the future.

## CONCLUSION

Preoperative serum CA724 might serve as a potential predictive biomarker for the prognosis of CRC patients with normal CEA levels. And more multicenter studies and statistics were needed to verify.

## ABBREVIATIONS

| | |
|---|---|
| **CRC** | colorectal cancer |
| **CEA** | carcinoembryonic antigen |
| **OS** | overall survival |
| **CA724** | carbohydrate antigen 724 |
| **CA199** | carbohydrate antigen 199 |
| **CA242** | carcinoembryonic antigen 242 |
| **CA125** | carcinoembryonic antigen 125 |
| **pT stage** | pathological tumor stage |
| **pN stage** | pathological node stage |

### Funding
This research was sponsored by Guangxi Self-financing Research Projects (Z20180959). The funders had no role in study design, data collection and analysis, decision to publish, or preparation of the manuscript.

### Grant Disclosures
The following grant information was disclosed by the authors:
Guangxi Self-financing Research Projects: Z20180959.

### Competing Interests
The authors declare that they have no competing interests.

## Author Contributions

- Jiaan Kuang performed the experiments, prepared figures and/or tables, and approved the final draft.
- Yizhen Gong performed the experiments, authored or reviewed drafts of the paper, and approved the final draft.
- Hailun Xie analyzed the data, prepared figures and/or tables, and approved the final draft.
- Ling Yan analyzed the data, prepared figures and/or tables, and approved the final draft.
- Shizhen Huang analyzed the data, prepared figures and/or tables, and approved the final draft.
- Feng Gao conceived and designed the experiments, authored or reviewed drafts of the paper, and approved the final draft.
- Shuangyi Tang performed the experiments, authored or reviewed drafts of the paper, and approved the final draft.
- Jialiang Gan conceived and designed the experiments, authored or reviewed drafts of the paper, and approved the final draft.

## Human Ethics

The following information was supplied relating to ethical approvals (i.e., approving body and any reference numbers):

The study protocol was approved by the Hospital Ethics Committee of the First Affiliated Hospital of Guangxi Medical University, Guangxi (approval number: 2018(KY-E-086)).

## Data Availability

Raw data is available in the Supplemental Files.

## Supplemental Information

Supplemental information for this article can be found online at http://dx.doi.org/10.7717/peerj.8936#supplemental-information.

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
