# Peer review of "The prognostic value of preoperative serum CA724 for CEA-normal colorectal cancer patients"

_PeerJ, doi:10.7717/peerj.8936_

## Round 0.1 · original submission · Major Revisions

Please pay attention to the reviewers suggestions particularly patient stratification and statistical analysis.

Reviewer 1 ·

Basic reporting

It is an interesting study focusing on the prognostic value of serum CA724 in CEA-normal colorectal cancer patients. It should be revised by a native English speaker and a professional editor. There are a lot of language and spelling errors in this study (see the attached pdf file). The figures and tables were poorly prepared and should be revised.

Experimental design

Experimental design is reasonable. More details of the inclusion and exclusion of patients should be added.

Validity of the findings

The findings were basically valid.

Additional comments

1. I suggest to use "elevated /normal CA724", not the "positive/negative CA724".
2. capitalize the first letter of a Word.
3. The conclusion is too affirmative. This is only a small retrospective study, the evidence level is low.
4. CA724 is not only a biomarker for gastric cancer, but also for gastrointestinal tract, lung, breast, and ovaries.
5. The defect of the study should be added to the DISCUSSION part.
6. The first Table shoulde be Table 1, not Table 2.
7. What's the clinical significance of Figure 3?

Annotated reviews are not available for download in order to protect the identity of reviewers who chose to remain anonymous.

Reviewer 2 ·

Basic reporting

It is an interesting study. In clinical practise, we meet many cases which CEA is low but CA724 is high.It is valuable to explore the preoperative CA724 with the prognosis for CRC.

Experimental design

It is ok , but I think caculate OS and DFS of other tumor mark than CA724 is not necessary.

Validity of the findings

Though CA724 is detected for patients in clinical practise for many years , such study is interesting and valuabl.

Additional comments

In abstract section, "the OS and DFS in patients with positive CA724 was poorer than those
with negative." "positive" and "negative" should altered to "high" and "normal",thus they may be consistent with the graphical description.The headline of Table 1 was written Table 2 by mistake.

Reviewer 3 ·

Basic reporting

The article seems good, but I recommend to send the paper to professional editing.
In the introduction section will be better to reduce the first paragraph because it contains too much general information. Also will be better to discuss more widely references 4 and 5. Line 52-I recommend paraphrasing it because the concentration of these markers can increase during other different diseases and pregnancy. Please also add information about CA 19-9 in combination with CEA.
Line 100 is empty, please check it.
It is impossible to understand Table 2. Please either split this table into 2 separate tables (OS and PFS) and separately show information about circulating markers.
In figure 2 please check commas in the title.

Experimental design

A research question is ok, but novelty is limited.
In methods section please provide information about tumor whether the patient was first diagnosed with a tumor or is it a relapse. Why the amount of metastatic patients is so small? Explain it, please.
Also in the abstract, you discuss the only R but use SPSS as well, It will be better to avoid this misunderstanding.
In the discussion section please consider hypermethylation of the Septin 9 gene as a diagnostic marker.
And is it possible to divide patients based on the left or right tumor localization?

Validity of the findings

All data seem reliable. The conclusion is robust but the discussion is not so detailed.

Reviewer 4 ·

Basic reporting

This manuscript is well written. Literature references are updated but novel information considering more recent publications on the European Group on Tumour Markers (EGTM) guidelines will enrich the introduction.

Experimental design

In this manuscript, authors evaluated the prognostic value of pre-operative serum levels of the carbohydrate antigen CA72.4 in colorectal cancer (CRC) patients with normal CEA levels. Therefore, a series of 295 colorectal cancer patients was retrospectively analysed and the serum preoperative levels of CA72.4, CA19.9, CA125 and CA242 were correlated with patients clinicopathological information, namely overall survival and disease-free survival. The authors have concluded that preoperative serum levels of CA72.4 may serve, in CRC, as a potential prognostic factor in patients with normal CEA levels.

Methods were described in a concise way but major methodologies are annoted. Ethical agreements were considered.

Validity of the findings

This study was interesting and focused on an important question. In fact, the serum presence of carbohydrate antigens have for a long time been investigated, aiming to identify novel biomarkers of disease progression and, eventually, of therapeutic response. Several attempts were performed in distinct cancer types. In CRC, in particular, the European Group on Tumour Markers (EGTM) guidelines clearly suggested the use of CEA as a prognosis marker, especially in stage II patients, as postoperative surveillance on stage II and III patients, but also to monitor therapy response in advanced disease. Nevertheless, patients with normal CEA levels (generally over 50%) have no other sufficiently strong prognostic serum marker. The use of CA19.9 and CA242 as a prognostic marker have not been consensually recommended.

Data is based on a retrospective study using a cohort of 295 patients. Statistical analysis may require some considerations and revision. Based on the analysis performed, conclusions are well stated but validation on another population would be certain an improvement, although not demanding.

Additional comments

This manuscript is well- written and analyses, in a retrospective way, a considerable cohort of 295 patients with complete clinicopathological information. Inclusion criteria and ethical considerations were well approached. The preoperative expression levels of the selected serum markers in patients who underwent surgery were assessed through a chemiluminescence immunoassay. The document is well ome considerations are taken in relation to the statistics.
Some points require however some considerations:
- The authors statistically analysed the expression of the distinct markers in relation to clinicopathological features using the Chi-square and not the Friedman’s test followed by intergroup comparisons with Wilcoxon test. The authors should better explain the reasoning that led to the choose of the selected statistical tests.
- It would be also interesting to indicate from the 295 patients that underwent surgical interventions to indicate those that had indeed relapsed.
- In this series reduced metastasis are accounted (approx. 3%) is there any possible explanation?
- It would be very important to stratify the cases not only in regard of TNM but also on clinical stage (I, II, III, IV) and to verify if any association with the expression of CA72.4 is emerging
- Knowing that colon tumors are different entities regarding their location, it would be also very important to stratify them in regard tumor location as left and right colon. Other emergent information in CRC is the immunoscore. Although this was not on the scope of the manuscript, it would be very interesting it authors could provide additional information, and discuss it in integration with the present findings, regarding the inflammatory infiltrate present, since many these carbohydrates antigens are also antigens for immune cells.

Minor comments:
1)table 1 is miss designated table 2. Please correct
2) the word “the” is repeated in the last sentence of the section survival follow-up

---

## Round 0.2 · Minor Revisions

The manuscript needs to be revised to improve the English language.

Reviewer 2 ·

Basic reporting

It has been improved after revision.

Experimental design

It is good design.

Validity of the findings

It is feasible.

Additional comments

It is an interesting study.

Reviewer 3 ·

Basic reporting

I still recommend professional editing and checking for errors.
Line 64 - references.
Line 79 - capital letters.
Line 95 and so on.
Also, I advise checking manuscript again.

Experimental design

There are no major changes since the previous submission.

Validity of the findings

Conclusions look reasonable.

---

## Round 0.3 · accepted · Accept

The manuscript is acceptable for publication.